# Antioxidant Capacity of Chitosan on Sorghum Plants under Salinity Stress

Takalani Mulaudzi [1,*], Mulisa Nkuna [1], Gershwin Sias [1], Ibrahima Zan Doumbia [1], Njagi Njomo [2] and Emmanuel Iwuoha [2]

1 Life Sciences Building, Department of Biotechnology, University of the Western Cape, Private Bag X17, Bellville 7535, South Africa
2 SensorLab, Department of Chemical Sciences, University of the Western Cape, Private Bag X17, Bellville 7535, South Africa
* Correspondence: tmulaudzi@uwc.ac.za

**Abstract:** Salinity stress is one of the major environmental constraints responsible for the reductions in agricultural productivity. Salinity affects crop growth, by causing osmotic and ionic stresses, which induce oxidative damage due to increased reactive oxygen species (ROS). Exogenous application of natural compounds can reduce the negative impacts of salinity stress on plants. This study evaluated the antioxidant capacity of chitosan, a biopolymer to reduce the salt-induced oxidative damage on sorghum plants. Morpho-physiological and biochemical attributes of sorghum plants stressed with 300 mM NaCl, in combination with chitosan (0.25 and 0.5 mg/mL), were assayed. Salt stress decreased growth, fresh (66.92%) and dry (48.26%) weights, affected the shape and size of the stomata, caused deformation of the xylem and phloem layers, and increased the $Na^+/K^+$ (1.3) and $Na^+/Si^+$ (5.4) ratios. However, chitosan effectively reversed these negative effects, as supported by decreased $Na^+/Si^+$ ratio (~0.9) and formed silica phytoliths. Oxidative stress was exerted as observed by increased $H_2O_2$ (44%) and malondialdehyde (125%) contents under salt stress, followed by their reduction in chitosan-treated sorghum plants. Salt increased proline (318.67%), total soluble sugars (44.69%), and activities of SOD (36.04%) and APX (131.58%), indicating sorghum's ROS scavenging capacity. The antioxidant capacity of chitosan was measured by determining its ability to reduce oxidative damage and minimizing the induction of the antioxidant defense system. Chitosan reduced oxidative stress markers, proline, total soluble sugars, and the antioxidant enzyme activities by more than 50%. Fourier Transform Infrared Spectra of chitosan-treated samples confirmed a reduction in the degradation of biomolecules, and this correlated with reduced oxidative stress. The results suggest that chitosan's antioxidant capacity to alleviate the effects of salt stress is related to its role in improving silicon accumulation in sorghum plants.

**Keywords:** anatomical structure; antioxidant; chitosan; FTIR; oxidative stress; ROS; silicon; *Sorghum bicolor*





## 1. Introduction

Salinity is one of the abiotic stresses that continues to affect agriculture by decreasing crop growth and yield globally [1]. About 7% of total land around the world and close to 20% of irrigated land areas are affected by salinity [2]. Moreover, in arid and semi-arid areas, manual or artificial irrigation is one of the ways to improve agricultural productivity [3]. However, poor irrigation, lack of rainfall, and other environmental factors are the main drivers of salinity increase in the ecosystem [4]. Salinity negatively affects plants by causing osmotic stress and ion toxicity, which lead to secondary stress and hence oxidative damage of cell membranes, and macromolecules such as lipids, proteins, and nucleic acids, leading to the deactivation of several cellular and metabolic processes [5–7]. Salinity-induced oxidative damage is primarily attributed to the excessive production of reactive

oxygen species (ROS) including singlet oxygen ($^1O_2$), superoxide radical ($O_2^{\bullet -}$), hydrogen peroxide ($H_2O_2$), and hydroxyl radical ($^{\bullet}HO$), which are regulated by the enzymatic and non-enzymatic antioxidants [8]. Salinity stress also causes nutrient imbalance, which affects photosynthesis, stomatal conductance, and ionic imbalance due to the accumulation of toxic ions ($Na^+$ and $Cl^-$), thus, hindering the absorption of essential elements (K, P, Mg, Si among others) [9]. Exclusion or compartmentalization of toxic $Na^+$, while lowering $Na^+/K^+$ or increasing $K^+/Na^+$ ratios [10,11], is one of the common strategies employed by plants to adapt under salt stress. Furthermore, salinity stress induces various biochemical changes within plant tissues, aiding in a positive gradient in osmotic potential between the soil and plant tissue for survival. This is commonly associated with the accumulation or loss of biomolecules including carbohydrates, lipids, or amines and the synthesis of organic molecules such as proline [12]. These changes can be rapidly detected by infrared spectroscopy, which proved vital in detecting infected leaves and stems of *Brassica napus* [13] and salinity-induced changes in *Calophyllum inophyllum* [12].

Plants also activate the production of osmolytes such as proline and soluble sugars to protect cells against the detrimental effects of salt stress through osmoregulation [13]. Proline also acts as a ROS scavenger in the form of a non-enzymatic antioxidant to counterbalance osmotic stress [14]. While the antioxidant defense system is one of the main ROS scavenging pathways, this system consumes energy, which results in halted growth to conserve energy for defense as part of stress tolerance in plants [15]. The negative effects of salinity stress and its impact on the enzymatic antioxidant scavenging defense system have been reported for many agricultural crops including *Vicia faba* [16], *Phaseolus vulgaris* [17,18], *Tricum aestivum* [19], *Ocimum basilicum* [20] *Lupinus termis* [21], *Zea mays* [22], as well as *Sorghum bicolor* [10,23], among others. The most investigated antioxidant enzymes, which act as ROS reducing agents under stress conditions include superoxide dismutase (SOD), peroxidase (POD), catalase (CAT), the ascorbate–glutathione (AsA-GSH) cycle enzymes such as ascorbate peroxidase (APX), monodehydroascorbate reductase (MDHAR), dehydroascorbate reductase (DHAR), glutathione reductase (GR), and glutathione peroxidases (GPX) [24,25]. There is a steady balance between ROS production and antioxidant defense systems within plant cells [24,26]. Furthermore, during unfavorable conditions, overproduction of ROS demolishes the equilibrium and causes cellular damage, resulting in programmed cell death as well as decreasing plant productivity [27].

The plant's epidermis forms part of the first line of defense against stress and plays an important role in water relation [28]. While the guard cells which form the stomata are located on the epidermal tissue, they mediate photosynthesis and transpiration rates. These microscopic gates are at the forefront in the exchange of carbon dioxide and water vapor in plants [29]. Guard cells regulate the stomatal aperture, and when NaCl accumulates in plants they induce a rapid closure of the stomata as observed in *Brassica napus* under salinity [30], among other plants and serve as a salt tolerance mechanism [31]. Due to the sensitivity of guard cells to various environmental changes such as humidity, $CO_2$, and light [29], stomatal conductance was shown to be heavily restricted in *Olea europaea* [32], *Saccharum officinarum* [33], and *Vigna radiata* [34] under various abiotic stresses.

As a reinforcement, plants, especially grass-like crops, often rely on external factors for increased adaptation [35]. Grasses may absorb nutrients such as silicon (Si) from their environment to form silica deposits on their epidermis, known as silica phytoliths, which lend support and structure due to their resilient nature [36,37]. Additionally, Si is known to play a role in ion compartmentalization, immobilization of toxic metals, and reduce oxidative stress [38,39] Although these silica microstructures positively influence stress tolerance in plants, abiotic stresses negatively impact their formation and thus compromise plant structure [40]. Their characteristics under abiotic stress remain unclear, however, a decrease in phytoliths was observed in *Triticum turgidum* under PEG-induced osmotic stress [40]. Therefore, stress management strategies that induce stress tolerance to plants without compromising growth are necessary. Different compounds with elicitor properties, which mitigate salt stress linked to plant defense mechanisms, have been

identified as bio-stimulants [41], among these compounds, chitosan and its derivatives deserve special attention.

　　　Chitosan (poly [1,4]-2-amino-2-deoxy-D-glucose) is a biopolymer which is produced from the deacetylation of chitin, obtained from fungi and the exoskeleton of crustaceans [42]. Chitosan has a unique structure, which is characterized by three functional groups including the amino group and primary and secondary hydroxyl groups that are responsible for enhancing its affinity [43]. Due to its excellent properties including non-toxicity, biodegradability, biocompatibility, and affordability, chitosan has been applied in several fields including the agricultural sector. It was first applied as a bio-stimulant in 1983 due to its ability to act as a proteinase inhibitor through the production of phytoalexin [44]. Since then, chitosan has been showing significant improvement in germination, growth, and flowering of different crop species such as cereals, fruits, and medicinal crops [45]. In addition to its antimicrobial activity, chitosan promoted germination parameters in *Begonia hiernalis* and *Zea mays* [46,47], encouraged early flowering in ornamental plants [48], and improved biomass (fresh and dry weights) in *Solanum tuberosum* [49]. The role of chitosan to improve salt stress tolerance has been reported in several plant species including *Lactuca sativa* L [13], *Glycine max* [50], *Zea mays* [51], *Triticum aestivum* [52], *Carthamus tinctorium* L. and *Helianthus annuus* L. [41,53], *Plantago ovata* [54], and *Vigna radiata* [55]. Moreover, chitosan's effectiveness in promoting salt tolerance might be due to improved water use efficiency, mineral nutrients uptake, photosynthesis, and reduced oxidative stress [13,56]. To date, only a few studies have reported on the application of chitosan in sorghum to improve yield [57], seed germination, and antifungal activity [58]. With all these interesting and significant characteristics of chitosan in agriculture, its mechanism of action and its role in alleviating salt stress in sorghum still remain elusive.

　　　Sorghum (*Sorghum bicolor* (L.) Moench) is the 5th most important cereal crop in the world after maize, rice, wheat, and barley and the 2nd in Africa after maize, serving as a staple food for both humans and animals [59]. Sorghum is a highly productive C4 photosynthetic crop that is moderately tolerant to drought and salinity, thus, it has great potential to serve as a model crop to investigate the mechanisms of stress tolerance in cereal crops [50,60–63]. However, sorghum is sensitive to high salt at early growth stages and longer salt exposure can limit early seedling establishment and reduce growth and yields [64]. Few studies have been conducted on the improvement in sorghum's tolerance to salt stress, using calcium and ZnO nanoparticles [10,23]. With the salinity predicted to affect ~50% of the arable land by 2050, several strategies are important to maintain sorghum growth, development, and yield under abiotic stresses. Thus, the current study elucidated the antioxidative capacity of chitosan to reduce ROS formation and prevent oxidative damage without affecting sorghium's normal growth.

## 2. Materials and Methods

### 2.1. Sorghum Germination and Growth Condition

　　　Sorghum (*Sorghum bicolor* (L.) *Moench*) seeds were purchased from Agricol, Brackenfell, Cape Town, South Africa. Sorghum seeds were prepared, germinated, and seedlings grown as described previously [23]. Briefly, seeds were decontaminated by firstly surface sterilizing with 70% ethanol for 5 min, followed by a 1 h incubation in 5% NaClO while shaking and subsequently washed using autoclaved double distilled water (ddH$_2$O). Seeds were then imbibed with autoclaved ddH$_2$O, while shaking overnight in the dark. After air-drying under the laminar flow, seeds were sown and then allowed to germinate on a sterile water-imbibed paper towel and placed in the growth chamber set at 25 °C with complete darkness. After 7 days, seedlings were transferred into potting (one seedling per pot) soil containing a mixture of double grow, all-purpose organic potting soil (bought from Stodels Garden Center, Eversdal Road, Bellville, Cape Town, SA) and vermiculite (2:1). Experimental pots were positioned in a complete randomized block design, divided into four groups (control; salt treatment; salt + 0.25 mg/mL chitosan; and salt + 0.5 mg/mL chitosan) and seedlings were allowed to grow in the greenhouse under controlled conditions

(25 °C/±3 °C day/night and 16 h/8 h dark regimes). Seedlings were treated following a method as explained by [64] with some modifications. Briefly, the pots ((size: 21 × 16 and 5 cm height) as shown in Figure 1A–D) containing seedlings were watered with about 100 mL of nutrient solution (Dr Fisher's Multifeed, 19:8:16 (43), Reg. No./Nr. K5293, Act No./Wet Nr. 36 of/van1947)), that was applied every second day for a week followed by watering with only distilled water for a week. On day 14 after planting, sorghum seedlings were irrigated with salt (300 mM NaCl) and chitosan (Sigma-Aldrich, (C3646-25G), isolated from shrimp shells, ≥75% deacetylated) solutions. Chitosan stock was prepared by dissolving in 0.1 M acetic acid, and further diluted to 0.25 and 0.5 mg/mL using ddH$_2$O. The irrigation was completed every second day for 1 week, after which plants were harvested on day 28 after sowing.

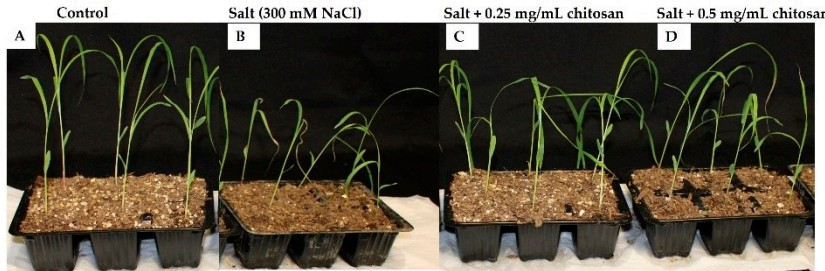

**Figure 1.** The phenotypic analysis of the positive effects of chitosan on the growth of sorghum under salt stress. Plants grown under (**A**) control, 0 mM NaCl; (**B**) salt: 300 mM NaCl; (**C**) salt + 0.25 mg/mL chitosan; and (**D**) salt + 0.5 mg/mL chitosan.

## 2.2. Growth Parameters

Growth attributes including plant phenotype, shoot length, fresh weights (FWs), and dry weights (DW) were measured as described previously [23]. Briefly, FWs were obtained by weighing fresh samples, on a Mettle Toledo AE50 analytical balance (Marshall Scientific, Hampton, US). DW were determined after oven-drying fresh samples at 55 °C or until a constant weight was attained.

The anatomic structure and the element distribution were also analyzed to observe the positive effects of chitosan on the growth of sorghum plants under salt stress. The anatomical structure (epidermis, xylem, and phloem) and element distribution of sorghum plants were analyzed at the University of Cape Town, South Africa, using High-Resolution Scanning Electron Microscopy (HRSEM) as described previously [23,65]. All spectra were analyzed using the built-in Oxford Inca software suite, while microphotographs were captured using the Tescan MIRA field emission gun scanning electron microscope operated at an acceleration voltage of 5 kV using an in-lens secondary electron detector.

## 2.3. Hydrogen Peroxide Content (H$_2$O$_2$)

Hydrogen peroxide (H$_2$O$_2$) was analyzed following an optimized method [66]. About 0.15 g of ground plant material were homogenized with 0.25 mL trichloroacetic acid (TCA), 0.5 mL potassium iodide (1 M), and 0.25 mL potassium phosphate buffer (10 mM, pH 6.0). Tubes were then vortexed and centrifuged for 15 min at 10,000 rpm (at 4 °C). Samples were transferred to 96 microwell plates and allowed to incubate at room temperature for 20 min. Absorbances were read at 390 nm using a FLUOstar® Omega (BMG LABTECH, Ortenberg, Germany) microtiter plate reader and H$_2$O$_2$ solution was measured by generating a standard curve.

## 2.4. Malondialdehyde Content (MDA)

Lipid peroxidation was determined by measuring malondialdehyde (MDA) formation following the thiobarbituric acid method as described by [67]. Fresh shoot samples (50 mg) were homogenized with 2 mL of 1% trichloroacetic acid (TCA (*w/v*)). The homogenate was centrifuged at 10,000 rpm (4 °C) for 10 min. Aliquots of 1 mL of the extract was added

to 2 mL of 20% TCA containing 0.5% thiobarbituric acid (TBA). Small holes were created in the cap of the 2 mL Eppendorf tubes using a syringe needle to prevent the tubes from bursting due to pressure from the heat. The mixture was boiled for 30 min at 95 °C and then allowed to cool on ice. The mixture was then centrifuged at 10,000 rpm for 15 min and the absorbance of the supernatant was measured at 532 nm using the FLUOstar® Omega (BMG LABTECH, Ortenberg, Germany), microtiter reader. The measurements were corrected for nonspecific turbidity by subtracting the absorbance at 660 nm.

### 2.5. Fourier-Transform Infrared Spectroscopy (FTIR) Analysis of Biomolecules

The FTIR spectrum of sorghum shoots was analyzed using a PerkinElmer Spectrum 100-PC FTIR Spectrometer (PerkinElmer (Pty) Ltd., Midrand, South Africa) as described by [23]. Samples were prepared using the KBr pellet method, where ~2 g of dry sorghum shoot tissue and 0.4 g of a pre-dried KBr were grounded in a mortar using a pestle to provide a homogeneous mixture. Then, the pellet mixture (~2 g) was scanned on a FTIR spectrometer. About 2 g of dry sorghum shoot tissues were analyzed on a wider spectral window between 450 and 4000 cm$^{-1}$.

### 2.6. Proline Content

The proline content was examined as described previously [68], with slight modifications. About 100 mg of ground sorghum shoots were re-suspended in 500 μL of 3% aqueous sulfosalicylic acid (3 g sulphuric acid (Mr = 218.185 g/mol), dissolved in 100 mL ddH$_2$O), followed by centrifugation at 13,000 rpm for 20 min. About 300 μL of the supernatant was mixed with 600 μL of 2.5% ninhydrin reaction mixture (1.25 g of ninhydrin dissolved in 30 mL of 99% acetic acid and 20 mL 6 M H$_3$PO$_4$) and boiled for 10 min in a water bath set at 95 °C. The sample was placed on ice and allowed to cool, and then equal volumes of toluene were added, and the optical density was measured at 520 nm using a FLUOstar® Omega (BMG LABTECH, Ortenberg, Germany) microtiter plate reader. The proline content was determined from a standard curve using pure proline as a standard.

### 2.7. Total Soluble Sugars

The total soluble sugars were determined as described previously [69] with some modifications. About 100 mg of grounded plant material were homogenized in 10 mL of ice-cold 80% acetone. The mixture was then centrifuged at 10,000 rpm for 10 min at 4 °C and 1 mL of the supernatant was added to a tube containing 3 mL of anthrone reagent (0.15 g anthrone, dissolved in 100 mL of 96% H$_2$SO$_4$). The samples were then placed in a boiling water bath set at 95 °C for 15 min, followed by a cooling reaction on ice until cold. The optical density was read at 620 nm using the Helios® Epsilon visible 8 nm bandwidth spectrophotometer (Thermo Fisher Scientific, Waltham, MA, USA). The total soluble sugar content was determined by generating a standard curve using glucose and the content was expressed as mg μg$^{-1}$ FW.

### 2.8. Enzyme Activity Assays

Samples for the determination of enzyme activities such as superoxide dismutase (SOD, EC, 1.15.1.11, BRENDA, The Comprehensive Enzyme Information System) and ascorbate peroxidase (APX, EC, 1.11.1.11, BRENDA, The Comprehensive Enzyme Information System) were prepared as previously described [70]. Plant material (0.5 g) was homogenized with 3 mL of 50 mM phosphate buffer (pH 7). The homogenate was filtered, followed by centrifugation at 18,000 rpm for 15 min using a refrigerated centrifuge set at 4 °C. The supernatant was stored at -20 °C until further assays were conducted.

#### 2.8.1. Superoxide Dismutase (SOD, EC, 1.15.1.11)

Total SOD activity was estimated by observing the reduction of photochemical of nitroblue tetra zolium (NBT) at 520 nm through a reaction mixture prepared as described [71]. Briefly, 1 mL of 75 μM riboflavin (0.283 g ribloflavin (CAS = 83−88−5) dissolved in 1

mL of $dH_2O$) was added into a 3 mL reaction mixture (12 mM methionine, 75 μL NBT, 50 mM potassium phosphate buffer (pH 7), 50 mM sodium carbonate ($Na_2CO_3$), and 0.1 mL enzyme extract). Plates were exposed to light for 20 min and absorbances were read at 560 nm using a FLUOstar® Omega (BMG LABTECH, Ortenberg, Germany) microtiter reader.

### 2.8.2. Ascorbate Peroxidase (APX, EC, 1.11.1.11)

Ascorbate peroxidase activity was assayed by estimating the decrease in optical density as a result of ascorbic acid at 290 nm. The reaction was prepared by mixing 50 mM potassium phosphate (pH 7), 0.1 mM EDTA, 0.5 mM ascorbate, 0.1 mL enzyme extract, and 0.1 mL of 0.1 mM $H_2O_2$ was added to initiate the reaction. The decrease in absorbance was estimated and measured at 290 nm using a FLUOstar® Omega (BMG LABTECH, Ortenberg, Germany) microtiter reader.

### 2.9. Statistical Analysis

All experiments (including plant growth and treatments and all assays) were repeated at least three times and data were statistically analyzed using a two-way ANOVA using GraphPad prism 9.2.0 (https://www.graphpad.com, by Dotmatics, accessed on 20 October 2021). The data in the figures and tables represent the mean ± standard deviation. Statistical significance between the control and treated plants was determined using Bonferroni's multiple comparison test and represented as **** = $p \leq 0.0001$, *** = $p \leq 0.001$, ** = $p \leq 0.01$, and * = $p \leq 0.05$. Pearson's correlation (r) matrix was calculated using the "GGally" and "mvtnorm" packages in R software.

## 3. Results

### 3.1. Chitosan Improves Sorghum Growth under Salt Stress

#### 3.1.1. Biomass

Salt stress severely affected sorghum growth as depicted in Figure 1 and Table 1. Salt caused a considerable reduction in sorghum growth and leaf expansion (Figure 1B) as compared to the control (Figure 1A). The phenotype of the chitosan-treated sorghum plants under salt stress showed growth improvement (Figure 1C,D). Furthermore, salt decreased shoot length (52%), fresh weight (66.9%), and dry weight (48.3%) of sorghum plants. However, exogenous application of chitosan improved shoot length by 33.9% (0.25 mg/mL chitosan) and 24.3% (0.5 mg/mL chitosan). Furthermore, FW and DW were restored by the exogenous application of chitosan, where 0.25 mg/mL chitosan increased FW by 79.5%, whereas 0.5 mg/mL chitosan increased FW by 30.9%, while DW increased by 53.02% (0.25 mg/mL chitosan) and 46.31% (0.5 mg/mL chitosan) in salt-stressed plants.

**Table 1.** Effects of chitosan on growth parameters of salt stressed *Sorghum bicolor*. Data represented are mean ± SD.

| Chitosan (mg/mL) | NaCl (mM) | Shoot Length (mm) | Fresh Weight (g) | Dry Weight (g) |
|---|---|---|---|---|
| 0 | 0 | 48.333 ± 6.506 | 2.920 ± 0.482 | 0.288 ± 0.045 |
| 0 | 300 | 24.667 ± 1.155 *** | 0.966 ± 0.040 **** | 0.1490 ± 0.003 ** |
| 0.25 | 300 | 33.00 ± 1.00 | 1.734 ± 0.142 | 0.228 ± 0.038 |
| 0.5 | 300 | 30.667 ± 1.15 | 1.265 ± 0.232 | 0.218 ± 0.083 |

Significant differences shown as **** = $p \leq 0.0001$, *** = $p \leq 0.001$, and ** = $p \leq 0.01$.

#### 3.1.2. Growth

To further determine the effects of chitosan on the growth and response of sorghum under salt stress, the anatomical structure (epidermis and vascular bundle) of sorghum plants was examined using Scanning Electron Microscopy (Figure 2). The epidermal tissue revealed the presence of stomata (circled in red; Figure 2A–D), while the vascular bundle revealed clear xylem and phloem tissues (Figure 2E–H). Salt stress caused a severe change

in the shape and size of the stomata (Figure 2B) as indicated by distinct and large guard cells (black arrows) in addition to increased stomatal aperture as compared to the control (Figure 2A). Treatment with chitosan reduced the size and aperture of the stomata under salt stress, with greater effects observed for high chitosan (0.5 mg/mL) concentration (Figure 3C,D). The tissues of the control (0 mM NaCl) sorghum plants (Figure 2E), showed smooth xylem and phloem layers, whereas that of salt-stressed (Figure 2B,F) plants showed rough, deformed, and shrunken layers as compared to the control. However, exogenous chitosan restored the vascular bundle (xylem and phloem) layers of sorghum under salt stress (Figure 2C,D,G,H), as observed by smooth layers as compared to plants treated with salt only. Additionally, a low concentration of chitosan (0.25 mg/mL) proved more effective in improving sorghum epidermis and vascular bundle tissue (Figure 2C,G).

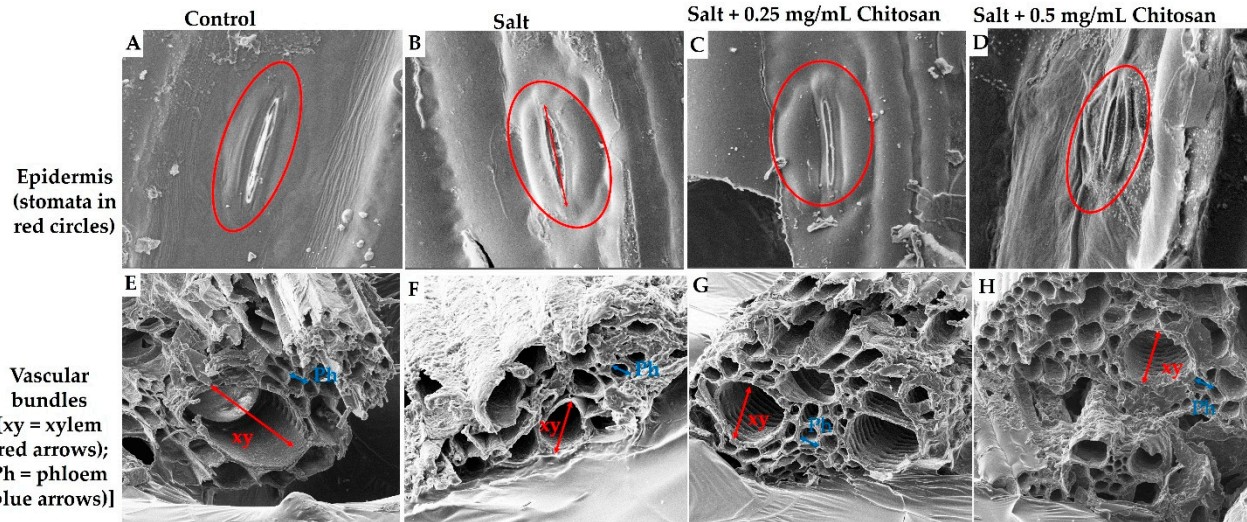

**Figure 2.** The positive effects of chitosan on the stomata (red circles, (**A**–**D**)) and vascular bundle layers (xylem = red arrows and phloem = blue arrows; (**E**–**H**)) of sorghum under salt stress. (**A**,**E**) (0 mM NaCl); (**B**,**F**) (300 mM NaCl); (**C**,**G**) (salt + 0.25 mg/mL chitosan); (**D**,**H**) (salt + 0.5 mg/mL chitosan).

### 3.1.3. Element Content Distribution

The distributions of sodium ($Na^+$), potassium ($K^+$), and silicon ($Si^+$) ions were analyzed using Scanning Electron Microscopy–Energy dispersive X-ray spectroscopy (SEM-EDX) in control samples and salt-treated samples in the absence and presence of chitosan (Table 2; Figure 3A–D). There was a high increase in $Na^+$ content (100%), whereas $K^+$ (51%) and $Si^+$ (47%) decreased under salt stress resulting in high $Na^+/K^+$ and $Na^+/Si^+$ ratios of 1.3 and 5.4, respectively. Treatment with chitosan increased all ions, $Na^+$ (2%), $K^+$ (21%), and $Si^+$ (499%), for 0.25 mg/mL chitosan, and $Na^+$ (31%), $K^+$ (16%), and $Si^+$ (747%) for 0.5 mg/mL chitosan, as compared to samples treated with salt only. This resulted in a further increase in the $Na^+/K^+$ ratio (1.5). Interestingly, chitosan caused a considerable reduction in the $Na^+/Si^+$ ratio from 5.4 (salt only) to 0.9 (0.25 mg/mL chitosan) and 0.8 (0.5 mg/mL chitosan), thus, showing an 85% decrease in $Na^+/Si^+$ ratio (Table 2; Figure 3A–D). These effects are clearly seen in the SEM-EDX spectra (Figure 3A–D).

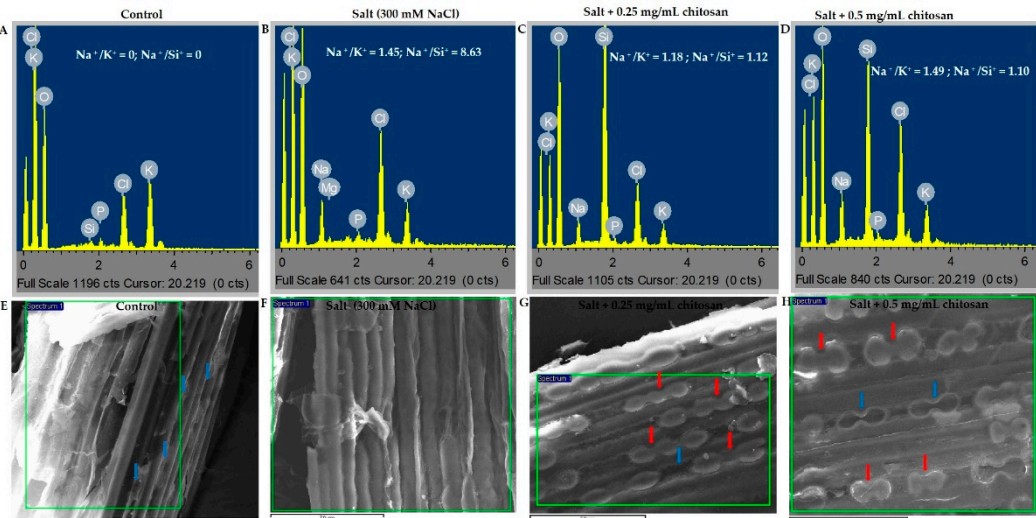

**Figure 3.** The effects of chitosan on the element distribution and formation of silica phytoliths on sorghum shoots under control and salt conditions. (**A–D**) SEM-EDX spectra showing the content of different ions, (**E–H**) SEM micrographs showing the sorghum surface for mapped ions, (the mapped area is shown by a green box). Silica cells and phytoliths are shown using blue and red arrows, respectively. (**A,E**) (0 mM NaCl); (**B,F**) (salt, 300 mM NaCl), (**C,G**) (salt + 0.25 mg/mL chitosan), (**D,H**) (salt + 0.5 mg/mL chitosan).

**Table 2.** Effects of chitosan on element distribution in sorghum shoots of control and salt stress conditions. Data represented are mean $\pm$ SD.

| Element | Control (Wt%) | Salt (Wt%) | Salt + CS1 (Wt%) | Salt + CS2 (Wt%) |
|---|---|---|---|---|
| $Na^+$ | 0.00 | $2.080 \pm 0.221$ * | $2.123 \pm 0.311$ | $2.740 \pm 1.222$ |
| $K^+$ | $3.15 \pm 0.522$ | $1.557 \pm 0.276$ * | $1.890 \pm 0.311$ | $1.807 \pm 0.743$ |
| Si | $0.743 \pm 0.713$ | $0.393 \pm 0.316$ | $2.357 \pm 1.384$ | $3.330 \pm 1.337$ * |
| **Element ratios** | | | | |
| $Na^+/K^+$ | 0.00 | $1.364 \pm 0.155$ *** | $1.520 \pm 0.341$ *** | $1.516 \pm 0.107$ *** |
| $Na^+/Si^+$ | 0.00 | $5.402 \pm 0.234$ **** | $0.976 \pm 0.602$ **** | $0.823 \pm 0.520$ **** |

Significant differences shown as **** = $p \leq 0.0001$, *** = $p \leq 0.001$, and * = $p \leq 0.05$.

The epidermal layers of the SEM-EDX investigated surface areas that represented smooth layers under control conditions (Figure 3E). Salt stress caused significant morphological changes depicted by shrinkage and deformation of the epidermal cell layers (Figure 3F). Treatment with chitosan improved the morphological structure of sorghum epidermis under salt stress, where additional structures called silica phytoliths were also observed in response to chitosan treatments (Figure 3G,H) under salt stress, which are associated with the absorption of silicon from the soil.

### 3.2. Chitosan Reduces ROS Formation and Membrane Damage on Sorghum under Salt Stress

#### 3.2.1. Hydrogen Peroxide and Lipid Peroxidation

To understand the level of salt-induced oxidative stress on sorghum plants and the ability of chitosan to ameliorate the effects of salt stress, the levels of $H_2O_2$ were measured (Figure 4A). The $H_2O_2$ content in sorghum plants treated with salt was 44% higher than the control plants. The exogenous application of chitosan (0.25 and 0.5 mg/mL) to salt-stressed plants significantly decreased $H_2O_2$ content by 52% in comparison to those treated with salt only

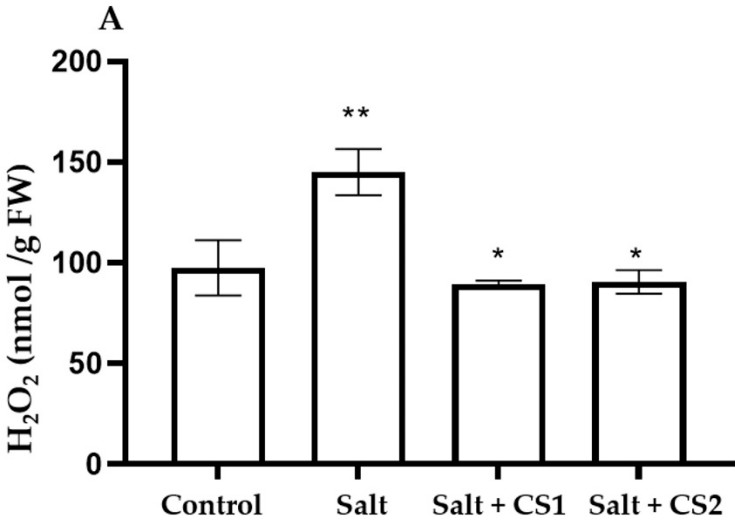

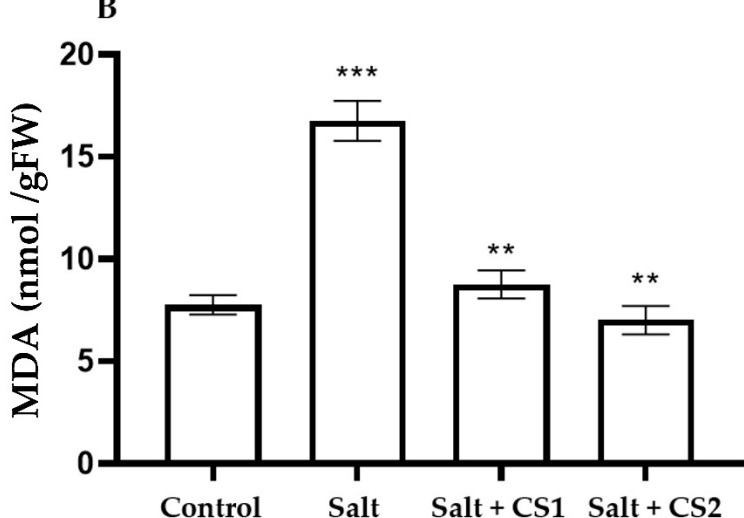

**Figure 4.** Chitosan (CS) reduces oxidative stress markers in sorghum under salt stress. (**A**) $H_2O_2$ and (**B**) MDA content in control (0 mM NaCl) and salt (300 mM NaCl) treated sorghum supplemented with chitosan (CS1 = 0.25 mg/mL and CS2 = 0.5 mg/mL). Error bars represent the SD calculated from three biological replicates. The statistical significance between control and treated seedlings was determined using a two-way ANOVA conducted on GraphPad Prism 9.2.0, shown as *** = $p \leq 0.001$, ** = $p \leq 0.01$, and * = $p \leq 0.05$.

To assess the degree of membrane damage by salt stress and the antioxidant ability of chitosan to prevent membrane damage, the study quantified MDA content, a byproduct of membrane lipid peroxidation (Figure 4B). Salt-treated sorghum plants showed a 50% increase in MDA content than the control. Interestingly, supplementing salt-stressed sorghum plants with chitosan (0.25 and 0.5 mg/mL) reduced MDA content by 50% as compared to plants treated with salt only.

### 3.2.2. Fourier Transform InfraRed Spectroscopic Analysis of Biomolecules

The nature and existence of biomolecules, such as phenolic compounds, proteins, carbohydrates, and lipids, were determined using Fourier Transform InfraRed (FTIR) Spectroscopy, as analyzed on a wider spectral region from 450 to 4000 cm$^{-1}$ (Figure 5). The infrared spectrum of the control sample (Figure 5A) showed a peak at 3525 cm$^{-1}$, which is found within the frequency range of 3600 to 3200 cm$^{-1}$, represented by O-H stretching

vibration, confirming the presence of phenolic compounds. Peaks at 2924[1] and 2835 cm$^{-1}$ are found under the frequency range of 3000 to 2850 cm$^{-1}$, represented by C-H stretching vibration of alkanes confirming the presence of aliphatic compounds (amino acids). Peaks 1589 and 1380 cm$^{-1}$ are within the 1400 to 1000 cm$^{-1}$ range and are represented by the C-F stretching vibration for the alkyl halide group, confirming the presence of carbohydrates. The spectra also showed the presence of alcohol, carboxylic acids, and fats (esters and ethers). These can be seen by a spectral peak at 1058 cm$^{-1}$, which is found within the 1320–1000 cm$^{-1}$ frequency range and shows a C-O stretching vibration. Peaks from 904 to 558 cm$^{-1}$ further confirm the presence of amino acids, representing the C-N and N-H stretching (Figure 5A) [41]. The FTIR spectrum of salt-stressed sorghum plants showed a huge shift in several peaks including a peak between 3525 and 2835 cm$^{-1}$, peaks at 1589 and 1380 cm$^{-1}$, and peaks from 904 to 558 cm$^{-1}$. However, exogenous chitosan, especially 0.25 mg/mL, partially restored the FTIR spectra of the salt-treated plants (Figure 5B, blue and green lines), bringing them closer to those of the control (Figure 5B, black line).

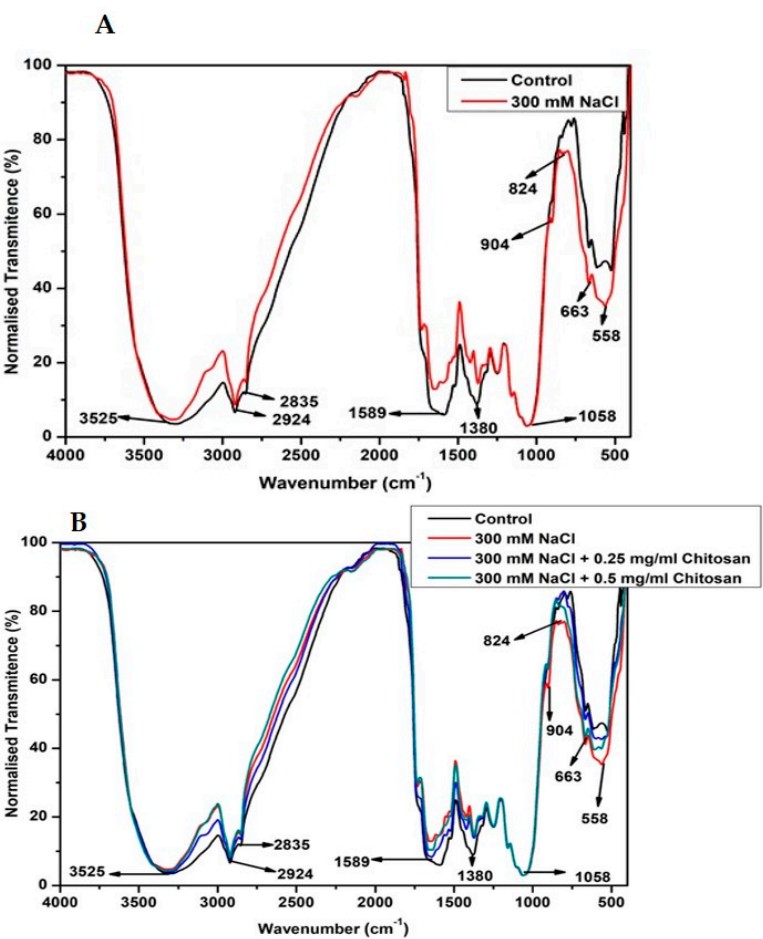

**Figure 5.** FTIR spectra analysis of the effect of chitosan on biomolecules in sorghum plants under salt stress. (**A**) Control (0 mM NaCl) represented as the black line and salt (300 mM NaCl) treated plants, represented as the red line, (**B**) salt-treated plants supplemented with 0.25 mg/mL (blue) and 0.5 mg/mL (green) chitosan.

*3.3. The Antioxidant Defense Effect of Chitosan on Sorghum under Salt Stress*

3.3.1. Proline and Total Soluble Sugars Content

The antioxidant effect of chitosan to regulate ROS detoxification was determined by firstly analyzing the accumulation of osmolytes including proline and total soluble sugars, and then determining the level of osmotic balance in salt-stressed sorghum plants (Figure 6A,B). Salt stress induced proline accumulation by 319% as compared to the control

(0 mM NaCl) (Figure 6A). Chitosan effectively induced stress tolerance by mediating a considerable decrease in proline content by more than 50% for both chitosan (0.25 and 0.5 mg/mL) concentrations under salt stress.

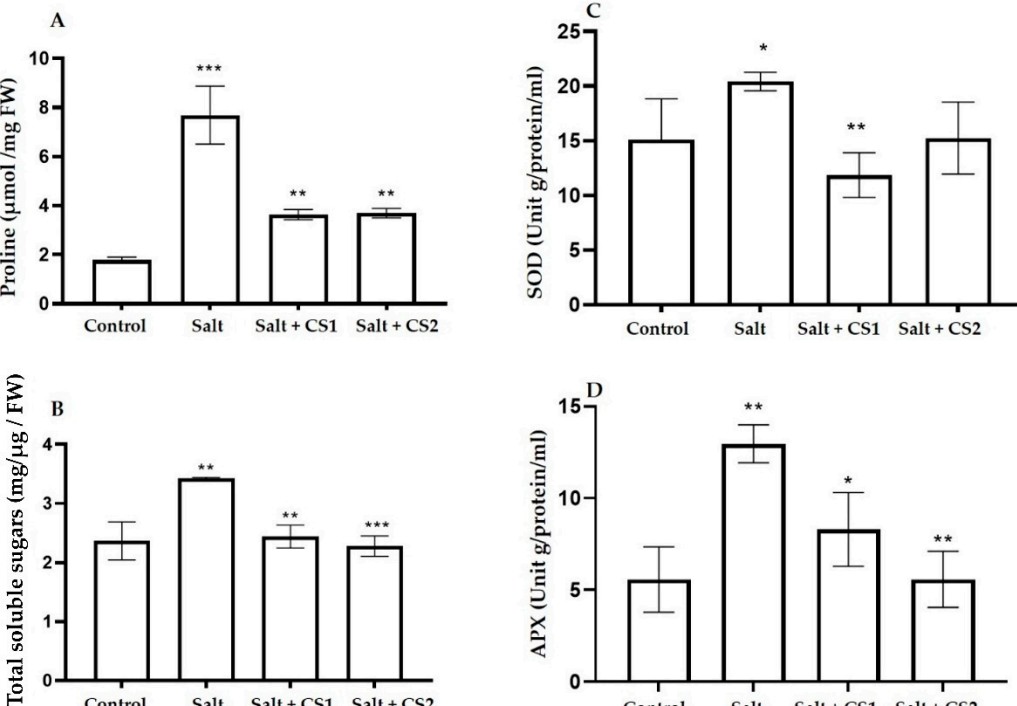

**Figure 6.** Exogenous chitosan (CS) reduced osmolyte accumulation and enzyme activities under salt stress in sorghum. (**A**) Proline content, (**B**) total soluble sugars content, (**C**) superoxide dismutase (SOD), and (**D**) ascorbate peroxidase (APX) in control (0 mM NaCl) and salt-treated (300 mM NaCl) plants supplemented with chitosan (CS1 = 0.25 mg/mL and CS2 = 0.5 mg/mL). Error bars represent the SD calculated from three biological replicates. Statistical significance between the control and treated seedlings was determined using a two-way ANOVA conducted on GraphPad Prism 9.2.0, shown as *** = $p \leq 0.001$, ** = $p \leq 0.01$, and * = $p \leq 0.05$.

Salt stress increased total soluble sugars by 68.33% more in sorghum plants as compared to the control (0 mM NaCl). Interestingly, the application of chitosan on salt-stressed sorghum plants reduced total soluble sugars by 28.76% (0.25 mg/mL chitosan) and 33.28% (0.5 mg/mL chitosan) as compared to plants treated with salt only.

### 3.3.2. SOD and APX Activities

Superoxide dismutase (SOD) is known as the first line of defense by directly dismutating $O_2^{\bullet-}$ to $H_2O_2$ and $H_2O$, whereas catalase (CAT) and ascorbate peroxidase (APX) are the main $H_2O_2$ scavengers [42]. To determine the antioxidant defense capacity of chitosan on sorghum in response to salt stress, the enzymatic activities of SOD and APX were analyzed (Figure 6C,D). Salt stress increased SOD activity by 36% as compared to control plants (Figure 6C). However, supplementing salt-stressed plants with 0.25 mg/mL chitosan showed significant decrease in SOD activity by 41.88%, but the effects of 0.5 mg/mL chitosan were not significant.

Salt stress increased APX activity by 132% as compared to control plants (Figure 6D). Enzyme activity was increased in salt-treated sorghum plants, but this activity was significantly decreased by exogenous chitosan to 35.97% (0.25 mg/mL chitosan) and 56.96% (0.5 mg/mL chitosan) under salt stress.

### 3.4. Correlation in Parameters

The relationship between the different studied traits was revealed using Pearson's correlation (Figure 7). The results indicated that most of the traits such as proline, $Na^+/Si^+$, APX, MDA, and total soluble sugars were strongly positively correlated to each other with a coefficient correlation value (r) close to 1. $Na^+$ and $Na^+/K^+$ were strongly associated with proline and $Si^+$ whereas $Na^+/Si^+$, APX, MDA (r = 0.07), and soluble sugars were positively correlated to $Na^+$ and $Na^+/K^+$. A strong positive correlation was observed between SOD and APX with r = 0.05. Highly strong negative (r = −1; deep brown color) to strong negative (r = 0.7; light brown color) Pearson's correlations were estimated between $K^+$ and all traits and the same trend was observed with $Si^+$.

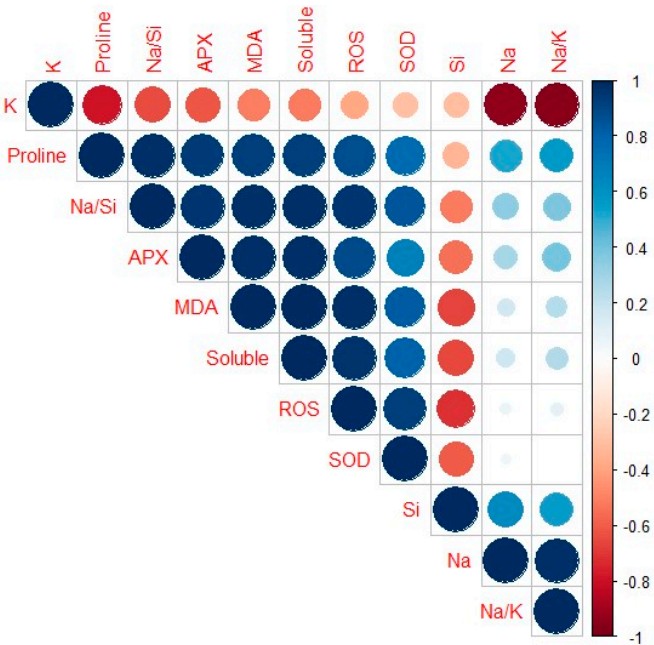

**Figure 7.** The relationship between different parameters in response to chitosan treatment under salt stress. ROS: reactive oxygen species ($H_2O_2$); MDA: malondialdehyde; Soluble: total soluble sugars; APX: ascorbate peroxidase; SOD: superoxide dismutase; Na: sodium ion; K: potassium ion, Si: silicon ion; Na/K: $Na^+/K^+$ ratio; Na/K: $Na^+/K^+$ ratio. The color of the circles and the coefficient values refer to the strength and significance of the correlation: 0.75–1 = strongly correlated; 0.5–0.75 = highly correlated; 0.25–0.50 = moderately correlated; and 0–0.25 = weekly correlated. Under 0 = negatively correlated.

## 4. Discussion

Salinity has been considered the most problematic factor in the agricultural sector since it reduces plant growth, development, and hence productivity [9]. Plants respond to such stress by slowing or hindering growth, among other traits [72]. In this study, 300 mM NaCl was chosen as a concentration to cause high salt stress on sorghum. Salt stress negatively affected sorghum growth and this was evident by reduced plant growth (Figure 1), shoot length, FW, and DW (Table 1). A reduction in biomass is indicative of growth limitation, thus, in this study growth reductions are attributed to the effects of osmotic stress, which interferes with metabolic processes, reducing the energy required for growth [73]. Similarly, growth reductions were observed in sorghum plants treated with 400 mM NaCl [23] and on other important plants such as *Schezonepeta tenuifolia* Briq [74], *Mentha pulegium* [75], *Cuminum cyminum* L. [76], and *Zea mays* [77]. Furthermore, the decline in biomass and growth might be due to the inhibition of cell expansion caused by low turgor pressure in salt-stressed plants resulting in a reduction in the shoot growth [78]. Due to its vast properties, including the mediation of growth and development and the responses to stress,

chitosan was chosen to study its antioxidant ability to eliminate the effects of stress on sorghum [79]. Exogenous chitosan reversed the effects of salt stress on sorghum growth resulting in improved phenotype, increased shoot length and biomass. This might be due to chitosan's role in improving physiological processes, such as cell division, stimulation of growth hormones, nutrient absorption and protein synthesis [80–83]. Exogenous chitosan was previously shown to overcome the negative effects of stress on growth as reported in *Zea mays* [84], *Triticum aestivum* [85], *Oryza sativa* [86], *Lactuca sativa* L. [13], *Silybum marianum* (L.) Gaertn. [81], and *Corchorus olitorius* L. [87].

The effect of chitosan on growth was also evaluated by analyzing the anatomic structure (stomata, xylem, and phloem) of sorghum plants under salt stress (Figure 2). Considerable reductions in plant growth were observed in salt-stressed sorghum plants, and this correlated with severe changes in the stomatal shape and size, as well as the xylem and the phloem layers suggesting that water loss was permitted through the salt-induced opened stomata (Figure 2B) whereas transport of water and nutrients through the xylem and phloem was affected (Figure 2F), hence, leading to decreased growth [88,89]. This is true since a high $Na^+$ concentration was measured under salt stress while there was a decrease in $K^+$ and $Si^+$ (Table 2; Figure 3B) as evident by the high $Na^+/K^+$ (1.3) and $Na^+/Si^+$ (5.4) ratios. Salinity also reduces the structure and density of the stomata, thus, altering its role for gaseous exchange and photosynthesis [90]. However, in this study, salt stress led to an increase in the density and aperture of the stomata, which is common in salt-tolerant plants [91]. The opening of the stomata in sorghum under salt stress could suggest $Na^+$ exclusion via the stomatal pores, thus, inducing tolerance but limiting growth. It was observed in this study that chitosan led to a decrease in stomatal size and hence closed the pores, and this correlated with improved xylem and phloem tissues and plant growth. It has been noted that chitosan is able to close the stomata [56,90], thus, inducing tolerance. Chitosan negatively affected the $Na^+/K^+$ ratio (1.5), which contradicted the improved growth under salt. Taken together, these results suggest that salt stress negatively affected the uptake and transport of nutrients (water and essential elements) by altering the structure of the epidermis, stomata, xylem, and phloem tissues.

Surprisingly, it was observed that instead chitosan increased the $Si^+$ distribution under salt stress, which correlated with improved growth and prevented damage of the anatomic structures, as seen by the closed stomata, improved surface layers, and wider openings of the xylem and phloem tubes, compensated by lowered $Na^+/Si^+$ (~0.9) ratios. Furthermore, chitosan treatment resulted in high levels of silica deposits as compared to the control (0 mM NaCl) and salt (300 mM NaCl) only (Figure 3E–H). These results suggest that chitosan-improved growth in sorghum was mediated by high silicon absorption, which resulted in the formation of silica phytoliths. This result further confirms that chitosan-improved growth under salt stress is strongly related to nutrient uptake [80,83]. The formation of silica phytoliths seems to have provided strength and support against epidermal and vascular bundle layer shrinkage and deformation, which prevented inhibition of the plant's metabolic processes [37,92]. To our knowledge. this is the first study to show that chitosan improved the anatomic structure (stomata, xylem, and phloem) and silica deposits of plants under salt stress (Figures 2 and 3).

This study chose to quantify $H_2O_2$ and MDA contents in order to understand the antioxidant ability of chitosan to reduce oxidative stress in sorghum (Figure 4). The results showed that $H_2O_2$ content (oxidative stress marker) was very high as compared to control plants, and hence this led to oxidative damage as measured by high MDA content, an indicator of membrane lipid peroxidation [93]. However, exogenous chitosan prevented ROS-induced oxidative stress in salt-treated sorghum plants by reducing $H_2O_2$ and MDA contents to the same magnitude as that of their controls. These results suggest the antioxidative power of chitosan to reduce oxidative damage and might suggest a role in directly scavenging ROS [45] or that the oxidative stress was reduced by the high content of accumulated silicon in response to chitosan treatment [38]. The positive effects of chitosan

reducing ROS-induced oxidative stress have been reported previously in lettuce [81], peppers [91], and durum wheat [94].

Salt stress also has the tendency to cause degradation and changes in biomolecules such as lipids, DNA, protein, and carbohydrates [95,96]. To further evaluate the antioxidant capacity of chitosan to prevent oxidative damage, the study used Fourier Transform Infrared Spectroscopy (Figure 5), a very important technique to reveal the different types of organic and inorganic compounds present in an organism [97]. Salt stress induced changes in the molecular component of sorghum plants by causing a major shift in several peaks that corresponds to the bonds forming phenols (3525 cm$^{-1}$), carbohydrates (1589 and 1380 cm$^{-1}$), lipids (1058 cm$^{-1}$), and amino acids (904–558 cm$^{-1}$) (Figure 5A), suggesting the induction of the non-enzymatic antioxidant defense system. However, exogenous chitosan at a low concentration of 0.25 mg/mL partially reversed the effects of salt stress (Figure 5B), further suggesting the antioxidant capacity of chitosan in scavenging ROS, hence, reducing oxidative damage.

In this study, sorghum plants accumulated high levels of toxic species (Figure 4), but plants have developed ROS scavenging systems including the non-enzymatic ones such as proline, flavonoids, and phenolic compounds, and the enzymatic systems such as SOD, APX, and CAT [98]. Proline is one of the well-researched osmolytes that plays a role in maintaining the osmotic balance by improving the osmotic potential during osmotic stress [99]. Proline accumulation is a common observation in response to abiotic stress in plants [100], suggesting a positive correlation between its accumulation and plant stress tolerance [101]. Consistent with the above, sorghum plants accumulated a high proline content under salt stress (Figure 6A), and this correlated with high ROS formation and MDA content. Based on these results, the study suggests that proline accumulation was triggered to scavenge ROS, maintain membrane stability, protect biomolecules from oxidative damage, and to control the osmotic balance and homeostasis [100,102]. Similarly, proline accumulation was observed previously in sorghum during germination [43] and vegetative [23,64,103] growth stages.

Plants also induced the synthesis of soluble sugars under salt stress as a tolerance mechanism [104]. There was a high content of total soluble sugars in salt-stressed sorghum plants (Figure 6B). Similar responses were observed in sorghum plants [23], and in *Lygeum spartum* [105], under salt stress. These findings further indicated that sorghum plants exhibited higher adaptive osmotic potential under salinity stress as evident by the high accumulation of proline and soluble sugars [106]. However, the application of chitosan on salt-stressed sorghum plants showed a positive effect by significantly decreasing both proline and the total soluble sugars to the same level as that of their control. These results are consistent with previous findings where chitosan application reduced osmolyte accumulation under salt stress such as in *Silybum marianum* L. [81] and *Glycine max* L. [107]. The ability of chitosan to mediate the reduction in osmolytes under salt stress is associated with the induction of tolerance through osmoregulation and osmoprotection [108], suggesting that chitosan also plays a structural role in plants. This is true since chitosan is isolated from the cell wall of fungi and the exoskeleton [40], thus, its role in providing strength and stability is relevant.

Antioxidant enzymes play a significant role in salt-stress tolerance through the detoxification of ROS and hence prevention of oxidation damage [109]. As the first line of enzymatic defense, SOD scavenges $O_2^{\bullet-}$ and dismutate it to $H_2O_2$ and $O_2$, followed by the detoxification of $H_2O_2$ by APX, CAT, and POD [110]. The results of this study showed that both SOD and APX activities (Figure 6C,D) were induced under salt stress, as compared to their controls, however, APX was induced to a greater extent than SOD. Furthermore, salt stress significantly increased antioxidant enzyme activity in other cereals such as *Triticum aestivum* [111], *Zea mays* [112], and *Oryza sativa* [108]. The increased activity of the antioxidant enzymes under salt stress indicates the level of ROS scavenging capacity mediated by sorghum. Their activities were reduced by exogenous chitosan, except for the high concentration (0.5 mg/mL chitosan), which showed no differences for SOD activity, but

both chitosan (0.25 and 0.5 mg/mL) concentrations reduced APX activity under salt stress. Although other reports have shown that chitosan increases osmolyte accumulation and antioxidant enzyme activities under salt stress [13,86], based on the results of this study, we suggest that the effects of chitosan under salt stress are differentially influenced by species type and whether the plant is tolerant or sensitive to salt. Furthermore, since sorghum is moderately salt-tolerant, the reduced osmolytes and enzymatic antioxidant activities under salt stress caused by exogenous chitosan might suggest that chitosan enhanced sorghum's tolerance and provided the antioxidant effect and osmotic regulation trait without sacrificing the energy meant for growth [71]. This study proved the positive effects of chitosan since its application in salt-stressed sorghum plants mediated tolerance by mainly lowering the $Na^+/Si^+$ ratio, the oxidative stress markers ($H_2O_2$ and MDA contents), and minimized activation of the defense systems, but improved uptake of essential elements (Si), suggesting chitosan's antioxidative ROS scavenging power. A strong positive correlation was observed (Figure 7) among all studied traits except for $Na^+$, $K^+$, $S^+$, and $Na^+/K^+$ in response to chitosan application under salt stress. The positive correlation maybe due to reduced oxidative damage due to chitosan's role in eliminating the accumulation of ROS, MDA, and other toxic ions [13,45], which correlated highly with reduced $Na^+/Si^+$ ratio, osmolyte accumulation [81,107,108,113], and antioxidant contents under salt stress. The more positive the correlation among traits is, the easier the transfer of traits together. The results from this study indicated that chitosan strongly influenced $Si^+$ accumulation and distribution under salt stress, where $Si^+$ strongly regulated the reduction of oxidative stress.

## 5. Conclusions

Although several studies suggested that the chitosan-mediated salt stress alleviation is related to the effective induction of the antioxidant system, this study added new knowledge, and suggested that a silicon-mediated ion homeostasis pathway exists that is activated by chitosan, however, these assumptions require further experimental confirmation. Furthermore, the results showed that both 0.25 and 05 mg/mL chitosan concentrations are effective and consistent for mediating the antioxidant capacity of chitosan on sorghum under 300 mM NaCl at the vegetative growth stage, without causing damage to plant cells. Therefore, future work to further affirm these findings is necessary, and this could include the measurement of the antioxidant scavenging radical activity of chitosan using 2,2'-azino-bis (3-ethylbenzothiazoline-6-sulfonate (ABTS) and 2,2-diphenyl-1-picrlylhyrazyl (DPPH) assays and to analyze the gene expression of the antioxidant enzymes and the $Na^+$-, $K^+$-, and $Si^+$-related transporters [10,11,105].

**Author Contributions:** Conceptualization, T.M.; methodology, M.N., G.S., N.N. and I.Z.D.; software, T.M., I.Z.D. and E.I.; validation, T.M., M.N., N.N., I.Z.D. and E.I.; formal analysis and investigation, M.N., N.N. and G.S.; resources, T.M. and E.I.; data curation, T.M.; writing—original draft preparation, M.N.; writing—review and editing, T.M., N.N. and E.I.; visualization, supervision, T.M., N.N. and E.I.; project administration, T.M. and E.I.; funding acquisition, T.M. All authors have read and agreed to the published version of the manuscript.

**Funding:** This research was funded by the National Research Foundation of South Africa: Thuthuka Institutional Funding (UID: 121939), grant.

**Institutional Review Board Statement:** Not applicable.

**Data Availability Statement:** Not applicable.

**Acknowledgments:** We would like to acknowledge the colleagues at the department of Biotechnology, UWC and the SensorLab at Chemical Sciences department, University of the Western Cape, for providing the space and most of the equipment that are situated in the laboratories available for use.

**Conflicts of Interest:** The authors declare no conflict of interest. The funders had no role in the design of the study; in the collection, analyses, or interpretation of data; in the writing of the manuscript, or in the decision to publish the results.

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
