# Peer review of "Antioxidant Capacity of Chitosan on Sorghum Plants under Salinity Stress"

_agriculture, doi:10.3390/agriculture12101544_

Round 1

Reviewer 1 Report

·         In the introduction part, a sufficient explanation of the role of chitosan in reducing salinity stress has not been mentioned well, and it is necessary to refer to the research of the last few years on this matter.

·         In the materials and methods section, it is not clearly specified where chitosan was obtained from. (Brand type and place of manufacture).

·         On what basis is the size of the pots chosen? The size of the pot is not suitable for this task according to the growth conditions of the investigated plant.

·         In the case of applying stress, additional information about the nutrients used should be mentioned.

·         How is salt stress applied?

·           Has the salinity level in the drainage water been measured in each irrigation?

·         In the case of measurement; the method of measuring the salinity of the soil and water drained from the pots should be stated.

·         How to apply the treatments is also debatable. Based on which reference did you do the foliar spraying?

·         Considering that chitosan foliar spraying has been done several times in a week and 28 days after the last foliar spraying, antioxidant activity and other traits have been measured, it may be only influenced by salinity stress and basically, the effects of chitosan in immediate reduction The effects are more noticeable after a week. What justification and explanation do you have?

·         Considering that the plant is used as a commercial aspect and has been under stress, additional information about biological performance and other similar traits are needed.

Author Response

Dear Reviewer1

Thank you for the thorough and great comments, that have improved our manuscript.

Below are the point-by-point response to your comments:

Comments from Reviewer 1, supported by the author’s point-by-point responses

  1. In the introduction part, a sufficient explanation of the role of chitosan in reducing salinity stress has not been mentioned well, and it is necessary to refer to the research of the last few years on this matter.

Author’s Reply: Our introduction was expanded and includes all relevant information explaining all key words from our title, and information regarding chitosan in reducing salt stress The role of chitosan to improve salt stress tolerance has been reported in several plant species including Lactuca sativa L [13], Glycine max [48], Zea mays [49], Triticum aestivum [50], Carthamus tinctorium L. and Helianthus annuus L. [51], Plantago ovata [52], Vigna radiata [53]. Moreover, chitosan’s effectiveness to promote salt tolerance might be due to improved water use efficiency, mineral nutrients uptake, photosynthesis and reduced oxidative stress [13; 54]. To date only a few studies reports on the application of chitosan in sorghum to improve yield [55], seed germination and antifungal activity [56]”

  1. In the materials and methods section, it is not clearly specified where chitosan was obtained from.

Author’s Reply: Information has been updated “chitosan was obtained from Sigma-Aldirich, (catalogue number: C3646-25G

  1. How is salt stress applied?

Author’s Reply: Salt was applied by irrigation, On day 14 after planting, sorghum seedlings were irrigated with Salt (300 mM NaCl) and chitosan [Sigma-Aldirich, (C3646-25G), isolated from shrimp shells, ≥75% deacetylated], Chitosan stock was prepared by dissolving in 0.1 M acetic acid, and further diluted to 0.25 mg/mL and 0.5 mg/mL using ddH2O”

  1. On what basis is the size of the pots chosen?

Author’s Reply: The study focused on sorghum plant at vegetative stage (V3 growth stage, also known as Stage 1), which occurs 10 to 15 days after emergence (as shown in Figure 1), we believe that the pot size was sufficient for this stage, unless if the focus was on other growth stages (V4, yield and grain).

  1. In the case of applying stress, additional information about the nutrients used should be mentioned

Author’s Reply: We have updated this information under section 2.1 “100 mL nutrient solution used “Dr Fisher’s Multifeed, 19:8:16 (43), Reg. No./Nr. K5293, Act No./Wet Nr. 36 of/van1947

  1. How is salt stress applied?

Author’s Reply: All treatments including salt stress were imposed by irrigating/watering (Nxele et al. 2017 (https://doi.org/10.1016/j.sajb.2016.11.003), with modification).

  1. Has the salinity level in the drainage water been measured in each irrigation? In the case of measurement; the method of measuring the salinity of the soil and water drained from the pots should be stated.

Author’s Reply: No! There was no need since the water in the drainage was not too much and the little that was there will normally evaporate within the same day of irrigation due to controlled greenhouse conditions, also based on the size of the pots used.

  1. How to apply the treatments is also debatable. Based on which reference did you do the foliar spraying?

Author’s Reply: In this study, treatment with salt and chitosan were done by watering/irrigation following method of Nxele et al. 2017 (https://doi.org/10.1016/j.sajb.2016.11.003), with some modifications.

Since several studies on chitosan application, used foliar spraying (Sheikhalipour et al., 2021, https://doi.org/10.3390/nano11030684; Salehi et al., 2017, https://doi: 10.18869/JHD.2017.101; Farouk and Amany, 2012, http://dx.doi.org/10.4314/ejb.v14i1.2, among others) and seed pre-treatment [Mahdavi and Rahimi, 2013, http://dx.doi.org/10.5053/ejobios.2013.7.0.9; Hameed et al., 2014 (Chitosan Seed Priming Improves Seed Germination and Seedling Growth in Wheat (Triticum aestivum L.) under Osmotic Stress Induced by Polyethylene Glycol, The Philippine Agricultural Scientist Vol. 97 No. 3, September 2014] and many others.

We opted for irrigation method to do something unique, in addition to growing at an average temperature of 25 reducing the effectivity of foliar application. The exposure time for stress treatment were done following the method of Zayed et al. 2017 (Effect of Nano chitosan on growth, physiological and biochemical parameters of Phaseolus vulgaris under salt stress, J. Plant Production, Mansoura Univ., Vol. 8 (5): 577-585, 2017), with slight modifications.

  1. Considering that chitosan foliar spraying has been done several times in a week and 28 days after the last foliar spraying, antioxidant activity and other traits have been measured, it may be only influenced by salinity stress and basically, the effects of chitosan in immediate reduction. The effects are more noticeable after a week. What justification and explanation do you have?

Author’s Reply: We would like to disagree with the reviewer on this note. If the effects on each trait were influenced by salinity only, then why did we observe the differences in the trait levels between the different treatments?. Furthermore, looking clearly at the anatomic traits, there is huge and unavoidable differences in the stomata, vascular bundles and epidermal layers of the control, salt and salt + chitosan treated samples.

  1. Considering that the plant is used as a commercial aspect and has been under stress, additional information about biological performance and other similar traits are needed.

Author’s reply: The scope of the current study was to evaluate the antioxidant capacity of chitosan to improve sorghum’s tolerance to salt stress at vegetative stage (v3), and we believe that the measured traits have provided those answers. The authors would appreciate if the reviewer can be specific about these traits, to be included in our future work. Based on the discoveries of the results, we have included some topics of interest in the conclusion.

Reviewer 2 Report

The reviewed manuscript entitled “Antioxidant capacity of chitosan on sorghum plants under salinity stress” is generally clear and well written, apart from lack of clarity in some parts, and some specifics. In this works, the authors showed that foliar application of chitosan can simulate of salinity stress of sorghum plants.

However, I have the following comments:

The novelty of the study needs to be highlighted compared to other similar studies.

The design of the experiment should be better explained

The method parts should be more descriptive.

The application of chitosan to improve plant resistance against salinity is promising, but the authors did not explain how to add chitosan to the plant by foliar spray? Or by irrigation?

Also, how to dissolve chitosan to get the solution? what solvent?

What is the concentration of chitosan solution used for each plant?

Also, has the soil and irrigation water used were analyzed?

Why did the authors rely on the salt concentration (300 mM NaCl)?

The results of this study depended largely on the anatomical responses, but there are no illustrative pictures of the autopsy and it was not well explained.

Anatomy images must be improved and the internal components of cells identified

From my point of view, the sodium content in plants should be measured to clarify the effect of salinity treatment on plant growth.

To show the effect on plant physiology, it is necessary to measure the change in photosynthetic pigments and also the effect on the rate of photosynthesis.

Figure 1 poor, should be improved for clarity

Discussion seems to be poor, didn't give good explanations of the results obtained. I think that it must be really improved. Where possible please discuss potential mechanisms behind your observations.

Discussion: Please describe the mechanism by which foliar application of a mixture of chitosan and organic acid affected various traits such as enzyme activity to improve salt tolerance.

 The discussion needs enhancement with real explanations not only agreements and disagreements. Authors should improve it by the demonstration of biochemical/physiological causes of obtained results. Instead of just justifying results, results should be interpreted, explained to appropriately elaborate inferences.

Author Response

Dear Reviewer 2

Thank you for the thorough and great comments, that have improved our manuscript.

Below are the point-by-point response to your comments:

Comments from Reviewer 2, supported by the author’s point-by-point responses

  1. The novelty of the study needs to be highlighted compared to other similar studies

Author’s Reply: The study is novel since this is the first study to investigate the antioxidant capacity of chitosan on salt-stressed Sorghum bicolor Moench L.

Even though some studies investigated the effect of chitosan on sorghum (Mohamed et al., 2020; Holguin-Pena et al., 2020) these studies investigated the effects of chitosan on yield, seed germination and antifungal activity under non-stress conditions.

Thus, as an important cereal crop ranking the 5th in the world and 2nd in Africa, it is important to investigate the different ways to improve its growth under abiotic stresses.

Several studies have investigated the effects of chitosan in alleviating the effects of salt stress in other important cereal crops, such as maize, wheat rice, etc. however our study is the first to show that chitosan positively improve growth by maintaining the anatomic (epidermis and vascular bundle) structure under salt stress, and to show its relationship with nutrient absorption. Additionally, we also showed that chitosan influences the absorption of silicon, which results in the formation of silica phytoliths (this is also novel) which are known to improve the structure and strength of plant tissues.

  1. The design of the experiment should be better explained

Reply: The missing information in the methodology section has been updated especially for section 2.1 [detailing on seed preparation, plant growth and treatment]. In section 2.7 we gave detailed methodology on how the antioxidant enzymes activity assays were done.

  1. The method parts should be more descriptive.

Author’s Reply: Methods were clearly elaborated on section 2.1, 2.7.1 and 2.7.2,

  1. How to dissolve chitosan to get the solution? What solvent?

Author’s Reply: This part was elaborated under section 2.1 “chitosan [Sigma, (catalogue number: C3646-25G), from shrimp shells, ≥75% deacetylated, 10 mg/mL stock solution prepared by dissolving chitosan in 0.1 M acetic acid, and further diluted with ddH2O to make 0.25 and 0.5 mg/mL working solutions]”. 

  1. What is the concentration of chitosan solution used for each plant?

Author’s Reply: Chitosan stock solution (10 mg/mL) was prepared by dissolving 10 mg of chitosan powder in 0.1 M Acetic acid. Then, to prepare working solution of 0.25 and 0.5 mg/mL, 10 mg/mL stock was further diluted using distilled H2O. Thus, plants were divided into six groups, [control; control + 0.25 mg/mL chitosan; control + 0.5 mg/mL (not included in this manuscript); salt treatment; salt + 0.25 mg/mL chitosan and salt + 0.5 mg/mL chitosan] were applied as mentioned in section 2.1.

  1. Has the soil and irrigation water used were analysed?

Author’s Reply: Soil and irrigation water (distilled water) used in this study were not analyzed. Potting soil “double grow, all-purpose organic potting soil” was obtained from Stodels (www.stodels.com), Cape Town, SA. This soil was used previously by other researchers [Nxele et al. 2017 (https://doi.org/10.1016/j.sajb.2016.11.003); Rakgotho et al., 2022, https://doi.org/10.3390/agriculture12050597]. We experienced no problems thus saw no need to send it for analysis.

  1. Why did the authors rely on the salt concentration (300 mM NaCl)?

Author’s Reply: Soils are regarded saline when the electrical conductivity (ECe) of a saturated paste soil extract is 4dS/m (40 mM NaCl) this is according to Munns (2008), and Sorghum bicolor (sweet sorghum) being more tolerant to salt stress as reported [Krishnamurthy et al., 2007; doi.org/10.1007/s10681-006-9343-9; Igartua et al., 1995; doi.org/10.1016/0378-4290(95)00018-L; Vasilakoglou et al., 2011; https://doi.org/10.1016/j.fcr.2010.08.011].

Based on these reports, and the continuous impact of the climate change on agriculture we see a need to conduct research with different salt levels, to identify the best stimulant to overcome high salt stress.

In the past we have treated sorghum with 100 mM NaCl, and observed less stress effects, thus it was not easy to test the positive effects of stimulants, since no significance was observed.

We decided to increase the strength of the salt, where 300 mM and 400 mM NaCl at vegetative growth stage 3-5 gave better results (Rakgotho et al., 2022 research article and MSc thesis), and 200 and 300 mM NaCl at germination stage (Mulaudzi et al., 2020; doi:10.3390/plants9060730] based on this findings we wanted to create an environment that is more saline but not too much to kill the plant.

  1. The results of this study depended largely on the anatomical responses, but there are no illustrative pictures of the autopsy and it was not well explained.

Author’s Reply: The anatomy figures have been improved: figures showing the plain epidermal tissues have been replaced with the epidermis showing the stomata (Figure 2) and silica phytoliths (Figure 3).

  1. Anatomy images must be improved and the internal components of cells identified.

Author’s Reply: The anatomy figures have been improved: cells (stomata), xylem and phloem are vascular bundle tissues that are made up of different cells, these tissues have been labelled in the figures (Figure 2). In addition, we added Figure 3, which shows the silica cells with or without silica phytoliths.

NB: note that scanning electron microscopy was used for these analysis, thus to be able to show the cells that make up these tissues we will require a different microscopy such as Transmission or confocal laser scanning microscopy, or any other microscopy.

  1. From my point of view, the sodium content in plants should be measured to clarify the effect of salinity treatment on plant growth.

Author’s Reply: Results on the element content have been provided (Table 2; Figure 3), originally these results were not included since a high chitosan concentration increased Na+ content, which affected the Na+/K+ ratio, however since we observed more interesting results that showed a positive influence of chitosan on silicon absorption, we decided to include them in the revised manuscript.

  1. To show the effect on plant physiology, it is necessary to measure the change in photosynthetic pigments and also the effect on the rate of photosynthesis.

Author’s Reply: This is true, however these assays where not measured in this study. Since the main focus for this study was on the anti-oxidative effect of chitosan on sorghum. On another note, measurement of osmolytes e.g Proline and Soluble sugars, can also explain the physiology, which are discussed in this manuscript. The physiology can also be explained based on the phenotypic analysis (Figure 1 & 2; Table 1) and anatomic structure showing the stomata and the silica phytoliths (Figure 3; Table 2).

  1. Figure 1 poor, should be improved for clarity

Author’s Reply: Figure 1, has been improved, this figure is now divided into two Figure 1, showing the phenotypic analysis and Figure 2, showing the anatomic analysis. Furthermore, several structures on the images are labelled.

  1. Discussion seems to be poor, didn't give good explanations of the results obtained. I think that it must be really improved. Where possible please discuss potential mechanisms behind your observations.

Author’s Reply: The discussion section has been revised.

  1. Discussion: Please describe the mechanism by which foliar application of a mixture of chitosan and organic acid affected various traits such as enzyme activity to improve salt tolerance.

Author’s Reply: We tried to deduce a mechanism in the revised manuscripts, based on the results that are presented in this manuscript, but without over-stating, more data is necessary to come up with the actual mechanism, as indicated in the conclusion.

  1. The discussion needs enhancement with real explanations not only agreements and disagreements. Authors should improve it by the demonstration of biochemical/physiological causes of obtained results. Instead of just justifying results, results should be interpreted, explained to appropriately elaborate inferences.

Author’s Reply: The discussion section has been revised. 

Reviewer 3 Report

Dear Authors

Please revise the manuscript according my comments

Regards

Author Response

Dear Reviewer 3

Thank you for your comments that have greatly improved our manuscript.

Below we provide a point-by-point responses to your comments.

Comments from Reviewer 3, supported by the author’s point-by-point responses

  1. Introduction

Author’s Reply: We are unable to see the comment.

  1. How did you apply salinity stress? Please more clarify

Author’s Reply: This comment is more detailed under reviewer 1’s answer: “salt was applied by irrigation”

  1. Please more explain about this method

A detailed information on how the FTIR analysis was conducted is provided.

  1. Please explain method in details for each enzyme (Enzymes)

Author’s Reply: The method section has been updated.

  1. The correlation table must be included.

Author’s Reply: At this stage we were not able to provide the correlation table, however the results provide are simple and clear.

  1. Explain how the plant tolerate through chitosan and include more discussion how are the physiological action performed?

Author’s Reply: The discussion has been updated.

Round 2

Reviewer 2 Report

The authors having responded to the suggestions and corrections requested, I therefore propose a publication in present form.

Author Response

Author’s reply: Authors are thankful for the time spent and effort made by the reviewer in evaluating our manuscript, which has greatly improved.

Reviewer 3 Report

Dear Authors

You did not provide correlation table and the methods were not complete. 

Author Response

Reviewer 3: You did not provide correlation table and the methods were not complete.

Author’s Reply: Authors are grateful to these comments, below is a point by point response.

  1. You did not provide correlation table:

Author’s Reply: Authors included a correlation figure (Figure 6).

  1. Methods were not complete

Author’s reply: Authors will appreciate if the reviewer can point out which method is still outstanding, however we included below a list of the methods that have been updated in the manuscript.

2.4 Fourier-Transform Infrared Spectroscopy (FTIR) analysis of biomolecules

FTIR spectrum of sorghum shoots was analysed using a PerkinElmer Spectrum 100-PC FTIR Spectrometer [PerkinElmer (Pty) Ltd., Midrand, South Africa] as described by [23]. Samples were prepared using the KBr pellet method, where ~2 g of dry sorghum shoot tissue and 0.4 g of a pre-dried KBr were grounded in a mortar using a pestle to give a homogeneous mixture. Then the pellet mixture (~2 g) was scanned on a FTIR spectrometer. About 2 g of dry sorghum shoot tissues where analyzed, on a wider spectral window between 450 to 4000 cm-1”.

Please explain method in details for each enzyme (Enzymes)

Author’s Reply: The method section has been updated. “2.7. Enzyme Activity assays

Samples for the determination of enzyme activities such as Superoxide dismutase (SOD, EC, 1.15.1.11) and Ascorbate peroxidase (APX, EC, 1.11.1.11) were prepared as previously described [69]. Plant material (0.5 g) was homogenised with 3 mL of 50 mM phosphate buffer (pH 7). The homogenate was filtered, followed by centrifugation at 18 000 rpm for 15 minutes using a refrigerated centrifuge set at 4 oC. The supernatant was stored at -20 oC until further assays were conducted.

2..7.1. Superoxide dismutase (SOD, EC, 1.15.1.11)

Total SOD activity was estimated by observing the reduction of photochemical of nitroblue tetra zolium (NBT) at 520 nm through a reaction mixture prepared as discribed [70]. Breifly, 1 mL of 75 µM riboflavin [0.283 g ribloflavin (CAS=83-88-5) dissolved in 1 mL of dH2O] was added into a 3 mL reaction mixture [12 mM methionine, 75 µL NBT, 50 mM potassium phosphate buffer (pH 7), 50 mM sodium carbonate (Na2CO3) and 0.1 mL enzyme extract. Plates were exposed to light for 20 minutes and absorbances were read at 560 nm using FLUOstar® Omega (BMG LABTECH, Ortenberg, Germany) microtiter reader.

2.7.2. Ascorbate peroxidase (APX, EC, 1.11.1.11)

Ascorbate peroxidase activity was assayed by estimating the decrease in optical density as a resultof ascorbic acid at 290 nm. Reaction was prepared by mixing 50 mM potassium phosphate (pH 7), 0.1 mM EDTA, 0.5 mM ascorbate, 0.1 mL enzyme extract and 0.1 mL of 0.1 mM H2O2 was added to initiate the reaction. The decrease of absorbance was estimated and measured at 290 nm using FLUOstar® Omega (BMG LABTECH, Ortenberg, Germany) microtiter reader.

  1. Explain how the plant tolerate through chitosan and include more discussion how are the physiological action performed?

Author’s Reply: The discussion has been updated “ We added more explanations of how chitosan mediate salt stress under each studied parameter in the discussion section”.